# Effectiveness of Motivational Interviewing in Regard to Activities of Daily Living and Motivation for Rehabilitation among Stroke Patients

**DOI:** 10.3390/ijerph17082755

**Published:** 2020-04-16

**Authors:** Hsiao-Mei Chen, Hsiao-Lu Lee, Fu-Chi Yang, Yi-Wen Chiu, Shu-Yuan Chao

**Affiliations:** 1Department of Nursing, Chung Shan Medical University, Taichung 40201, Taiwan; fiona@csmu.edu.tw (H.-M.C.); bethchiu@csmu.edu.tw (Y.-W.C.); 2Department of Nursing, Yuhing Junior College of Health Care & Management, Kaohsiung 80776, Taiwan; smart@ms.yuhing.edu.tw; 3College of General Education, National Chin-Yi University of Technology, Taichung 40201, Taiwan; caring686868@gmail.com; 4Institute of Allied Health Sciences, College of Medicine, National Cheng Kung University, Tainan 70101, Taiwan; 5Department of Nursing, Hungkuang University, Taichung 43302, Taiwan

**Keywords:** motivational interviewing, stroke, activities of daily living, rehabilitation motivation

## Abstract

Background: Stroke patients urgently need rehabilitation to enhance activities of daily living. This study aims to determine whether motivational interviewing (MI) improves the performance of activities of daily living and enhances motivation for rehabilitation among first-stroke patients. Methods: A quasi-experimental design was used in this study. The study recruited 65 patients between March and October 2016. Before the intervention, all patients received routine care. The experimental group (n = 33) received weekly sessions of MI for 6 weeks, whereas the control group (n = 32) received individual attention from a research nurse weekly for 6 weeks. Structured questionnaires were used to collect data, including demographic data, activities of daily living data (Barthel index {BI} and instrumental activities of daily living {IADLs} scale), and rehabilitation motivation data. Results: The BI and IADLs scores significantly improved with time in both the experimental and control groups. The generalized estimating equation approach showed that at 6 weeks and 3 months after the intervention, the rehabilitation motivation scores in the experimental group were respectively 3.10 and 2.54 points higher than those in the control group, with significant differences. Conclusions: MI could effectively enhance motivation for rehabilitation among stroke patients.

## 1. Introduction

Stroke is a major cause of acquired disability in the global population, and it places an increasing burden on healthcare resources [1]. According to the statistics of the World Health Organization (WHO), approximately 15 million people experience a cerebral vascular accident every year globally, and approximately 5 million people die of severe stroke [2]. The disability that affects activities of daily living after stroke can be measured using the Barthel index (BI) [3] and instrumental activities of daily living (IADLs) scale [4]. The BI is considered to be the basic performance index for measuring activities of daily living among stroke patients [3,4]. It has been found that self-care tasks of first-time stroke patients are affected and that mobility (walking) and stair climbing are the most affected [4]. Because of the limitations with regard to physical activity, stroke patients often sit for a long time (75%), and only 0.5% of stroke patients perform moderate-to-severe physical activities and 1.3 h of IADLs every day [5]. Patients should receive long-term and continuous rehabilitation therapy to reduce disability and dysfunction to recover activities of daily living [5,6]. Studies have shown that the greatest difficulty for stroke patients is facing changes in activities of daily living, and early rehabilitation therapy has been shown to significantly improve activities of daily living (BI and IADLs) among acute stroke patients with limb hemiplegia [3,4,6]. However, patients can become bored with rehabilitation after stroke, leading to a lack of rehabilitation motivation and, therefore, rehabilitation can be a challenging and extremely long and painful process for stroke patients [6]. It has been shown in related research that good rehabilitation motivation could enhance psychological function and confidence as well as the abilities required to deal with the after-effects of stroke, which would in turn be conducive to good stroke prognosis [7]. Therefore, when a professional team sets rehabilitation goals for an individual stroke patient, in addition to the provision of rehabilitation to the affected parts of the body, there is a need to observe the patient’s initiating role in terms of participation in rehabilitation [8]. Motivational interviewing (MI) is defined as a patient-centered, non-confronting communication method that applies five important principles of consultation, including rolling with resistance, expressing empathy, avoiding argumentation, developing discrepancy, and supporting self-efficacy, which is abbreviated as READS [9]. MI could improve the survival rate, medication compliance, and knowledge of stroke patients, change their lifestyle, and enhance their intrinsic motivation [10]. Currently, there is insufficient evidence of whether the MI intervention in first-stroke patients could influence their activities of daily living and rehabilitation motivation. The application of the BI, IADLs scale, and rehabilitation motivation scale for the assessment of stroke patients is helpful for understanding the recovery of physical function in these patients after MI, indicating their significance clinically [3,4,11].

The present study aimed to determine whether MI could improve the performance of activities of daily living (BI and IADLs) and enhance motivation for rehabilitation among first-stroke patients. It is hoped that stroke patients will actively learn to be independent with regard to daily activities and participate in rehabilitation to reduce the risk of stroke recurrence.

## 2. Materials and Methods

### 2.1. Study Design

This study applied a quasi-experimental design, including pre- and post-tests of experimental and control groups. Convenience sampling was used to recruit first-stroke patients from among stroke patients in the rehabilitation ward of a regional hospital in Taichung, registered in the Chinese Clinical Trial Registry, registration number ChiCTR1900024058. All participants received routine care and completed three questionnaires before the intervention (pre-test) and at 6 weeks (post-test 1) and 3 months (post-test 2) after the intervention. The experimental group received the intervention of individual face-to-face MI for 15–30 min per session once a week for 6 weeks in total. Conversely, the control group received individual attention from a researcher (nurse) at the bedside for 15–30 min per session once a week for 6 weeks in total. The researcher mainly provided the control group with stroke-related information, as well as care and greetings, and there was no attempt to enhance patient motivation or change patient behavior. The flow diagram of the intervention process is shown in Figure 1.

### 2.2. Participants

Physicians in the rehabilitation department referred first-stroke patients who met the inclusion criteria to the researchers, and these patients were divided into the experimental group and control group. Cluster assignment was adopted to avoid any contamination between the experimental and control groups, and the recruitment of the experimental group was completed first, followed by the recruitment of the control group. From March to October 2016, a convenience sample of 65 eligible stroke patients was recruited. The inclusion criteria were aged 18 years or above, diagnosis of first stroke within 3 months, ability to communicate in Mandarin Chinese or Taiwanese, without reading or dictation disabilities, and willingness to participate in this study and fill out an informed consent form. The exclusion criteria were a history of depression or nervous system diseases, such as Parkinson’s disease, psychiatric disease, and multiple sclerosis, previous treatment with a psychiatric or clinical psychology intervention, and inability to participate in questionnaire interviews.

G*Power [12] was designed as a general stand-alone power analysis program for statistical tests commonly used in social and behavioral research. The statistical software G power 3.0.10 (Heinrich-Heine-Universität Düsseldorf, Dusseldrof, Germany [12] was used to determine the sample size, with the alpha value set at 0.05, power at 0.8, and effect size at 0.70, according to the results of the study by Byers et al. [13] In addition, considering loss rates of 10% for refusal and 10% for withdrawal from the study [14], the total sample size required was 64. The initial sample size of this study was 72 participants. However, the final sample size was 65 participants, with 33 in the experimental group and 32 in the control group. During the study, three participants in the experimental group had depression (The full score of Center for Epidemiologic Studies Depression Scale (CES-D-10) was 30 points. Depression can be defined when CESD ≥10) and four in the control group did not complete the questionnaires 6 weeks after the intervention. These participants were excluded from this study, resulting in an effective acceptance rate of 90.2%.

### 2.3. Interventions

The MI intervention in this study was performed by a researcher with approximately 20 years of experience in the internal medicine and surgical departments. Before this study, the researcher took a three-credit course entitled, “Studies in Motivational Psychology” and completed 16 h of MI training and actual practice. The study was conducted by referring to the planning of an MI intervention to guide smokers in the study by Ahluwalia et al. [15] Evaluation of MI was conducted once a week for six weeks among the stroke patients in the experimental group based on the MI Assessment Form and in accordance with the operational manual for the MI for stroke patients for the assessment of motivation for change. The interview questions included “What behavioral changes did you experience in daily life after the stroke?,” “What is the duration of the change action?,” and “Why is there no willingness/action to enhance the care plan after the stroke?.” The stages of change were pre-contemplation, contemplation, preparation, action, maintenance, and termination, and the researcher assessed the stage of each stroke patient every week, according to behavioral characteristics [16,17]. The interventions of individualized behavior change strategies were applied according to the stage of change of the patient. The behavioral characteristics of the six stages were integrated with 10 behavior change strategies introduced by scholars. From the pre-contemplation stage to the contemplation stage, the strategy of consciousness raising was adopted; from the contemplation stage to the preparation stage, the strategies of dramatic relief and self-re-evaluation were adopted; from the preparation stage to the action stage, the strategy of self-commitment was adopted; and from the action stage to the maintenance stage, the strategies of reinforcement management, environmental re-evaluation, helping relationships, and stimulus control were adopted, allowing patients to transition from one stage to another successfully [16]. In addition, the 6-week MI intervention was conducted by using five principles of MI [17] and 10 basic therapeutic techniques, including asking permission, eliciting/evoking change talk, exploring importance and confidence, asking open questions, reflective listening, normalizing, decisional balancing, affirming, providing advice/feedback, and summarizing [17,18].

### 2.4. Measurement

Structured questionnaires were applied for data collection in this study, and the content was divided into the following three parts: (1) basic demographic information; (2) activities of daily living assessment (BI and IADLs scale); and (3) rehabilitation motivation assessment (rehabilitation motivation scale). Basic demographic information included sex, age, marital status, living conditions, number of people living together, religious beliefs, education level, current employment, and disease characteristics (number of diseases, smoking history, stroke type, brain area affected by stroke, hemiplegia, risk factors, etc.). Activities of daily living assessment included the BI and IADLs scale. The BI has been widely used in domestic and international studies [18,19]. The BI items include feeding, bathing, grooming, dressing, bladder management, bowel management, toilet use, bed/chair transfer, walking, and stair use, with the score ranging from 0 to 100 points, and good internal consistency was achieved when Cronbach’s α was at 0.90 [18]. The IADLs scale items include mode of transportation, shopping, laundry, food preparation, ability to use the telephone, housekeeping, responsibility for own medicines, and ability to handle finances, with each item having a score of 2–4 points (total score of 24 points). A lower score indicates a worse ability to perform tasks, whereas a higher score indicates a better ability to perform tasks and live alone. The re-test reliability in the literature was 0.93 [19], whereas Cronbach’s α in the present study was 0.86. The rehabilitation motivation scale was designed by Litman [20], and the purpose of this scale was to measure the motivation for rehabilitation of stroke patients. In this scale, self-assessment of the behavior to participate in physical therapy was made by the patients for the measurement of their motivational strength. The scale items include whether the patient is willing to undergo rehabilitation, cooperates in the rehabilitation process, tries hard for rehabilitation, actively requires or asks for resources, complains about the poor effectiveness of rehabilitation and about pain, excessively requires encouragement, and makes excuses to avoid rehabilitation. This scale uses a 4-point Likert scale, and the score ranges from 8 to 32 points, with an interrater reliability coefficient of 0.648. In February 2016, authorization and informed consent were obtained from the original author Guo Nai-Wen, and the Chinese version of the rehabilitation motivation scale for stroke patients was used in this study. Reliability analysis of the score of the 8-item rehabilitation motivation scale was conducted, and the results showed Cronbach’s α of 0.85, indicating intrinsic consistency.

### 2.5. Data Collection

“All subjects gave their informed consent for inclusion before they participated in the study. The study was conducted in accordance with the Declaration of Helsinki, and the protocol was approved by the Ethics Committee of K Hospital (approval number: REN 10332)”. Data collection was started after obtaining informed consent from the participants. A pilot study was initially conducted in February 2016, before formal data collection, and five stroke patients were recruited to fill out the questionnaires and engage in one session of MI. The purpose of this pilot study was to revise the questionnaires and operational manual for MI and to examine the reliability and validity of the instruments. To avoid any threat to the internal validity of the study, a double-blind procedure was adopted in data collection and the research assistants and patients were not aware of group assignment to prevent preconceived notions from affecting the accuracy of the study. The data collection process included assessment of eligibility, official recruitment, pre-tests in the experimental and control groups, post-tests in both groups at 6 weeks after the intervention, and post-tests via telephone interview in both groups 3 months after the intervention. The research assistants conducted a face-to-face questionnaire survey in the pre-test and the first post-test 6 weeks after the intervention in both the experimental and control groups. The collection of data 3 months after the intervention was conducted among the discharged patients using the same questionnaire via telephone by a trained research assistant who was supervised by the researcher to ensure the accuracy of the results.

### 2.6. Statistical Analyses

In this study, data were collected via a face-to-face questionnaire survey, and proofreading, coding, and data entry were conducted for valid questionnaires. Descriptive and inferential statistical analyses of the collected data were performed by using SPSS for Window 25.0 (IBM Corp., Armonk, NY, USA). Frequency distribution, percentage, mean, and standard deviation were assessed, and the homogeneity test, independent *t*-test, repeated measures ANOVA, and generalized estimating equation (GEE) approach were used. The homogeneity test was performed for the baseline data of the stroke patients in both the experimental and control groups. The chi-square test for contingency tables was performed for categorical data, and the independent-sample *t*-test was performed for homogeneity testing of continuous data. The independent-sample *t*-test was performed for pre-test BI, IADLs scale, and rehabilitation motivation scale data to determine differences in the pre-test data between the experimental and control groups. Repeated measures ANOVA was performed to analyze changes in BI, IADLs scale, and rehabilitation motivation scale data in both the experimental and control groups before the intervention and 6 weeks and 3 months after the intervention. The GEE approach was adopted to make an estimate in both the experimental and control groups to understand the effectiveness of the MI intervention in stroke patients (experimental group) and in those who received individual attention (control group). For all statistical analyses, a *p*-value < 0.05 was considered statistically significant.

## 3. Results

### 3.1. Basic Demographic Attributes

The study included 65 participants, with 33 in the experimental group and 32 in the control group. The mean (standard deviation [SD]) age in the experimental group was 61.61 (14.71) years, whereas that in the control group was 61.31 (2.60) years. With regard to activities of daily living, the mean (SD) BI score was 36.56 (2.15) points, reaching severe dependency (range from 21 to 60 points), whereas the mean IADLs score was 4.81 (0.37) points, indicating a relatively low ability to live alone. Additionally, the mean rehabilitation motivation score was 21.37 (3.40) points. On analyzing the differences in relevant variables of basic attributes and disease characteristics between the experimental and control groups before the intervention using the *t*-test and chi-square test, no significant differences were noted between the two groups (Table 1).

### 3.2. Changes in Activities of Daily Living (Barthel Index, BI, and Instrumental Activities of Daily Living, IADLs) of Stroke Patients After the Motivational Interviewing (MI) Intervention

The study compared the changes in activities of daily living (BI and IADLs) before and 6 weeks and 3 months after the MI intervention, as well as the differences in the results after the intervention. In the experimental group, the repeated measures results showed that the mean (SD) BI scores before (T1) and 6 weeks (T2) and 3 months (T3) after the MI intervention were 32.66 (16.66), 51.20 (20.67), and 59.38 (23.51) points, respectively. The mean BI score showed an increasing trend over time, with a significant difference (*F* = 42.90, *p* < 0.001). The mean deviations (MDs) for T2 − T1 and T3 − T1 indicated significant increases, with values of 18.59 and 26.72 points, respectively (both *p* < 0.05) (Table 2).

In the control group, the repeated measures results showed that the mean (SD) BI scores before and 6 weeks and 3 months after the control intervention were 40.45 (17.96), 58.03 (21.50), and 63.15 (26.63) points, respectively. The mean BI score showed an increasing trend over time, with a significant difference (F = 30.85, *p* < 0.001). The MDs for T2-T1 and T3–T1 indicated significant increases, with values of 17.58 and 22.70 points, respectively (both *p* < 0.05) (Table 2).

On comparing the mean BI score before and 3 months after the interventions in the two groups, it was found that the score increased by 26.72 points in the experimental group and increased by 22.70 points in the control group after the intervention. In both groups, the BI scores were higher after the intervention (both time points) than before the intervention, with significant differences (both *p* < 0.05) (Table 2). The GEE approach was used for the difference in BI scores before and after the interventions in the two groups, and no significant difference was noted (*p =* 0.436), but a significant difference was noted with regard to the time effect (*p <* 0.001). Thus, the mean BI score increased significantly with time, but it did not show a significant difference with regard to the MI intervention (Table 3).

The repeated measures results of the IADLs scale showed that the scores before and 6 weeks and 3 months after both interventions kept increasing, with mean (SD) IADLs scores of 4.38 (3.21), 6.84 (3.82), and 7.56 (3.78) points, respectively, in the experimental group and 5.24 (2.70), 8.48 (3.95), and 9.39 (5.03) points, respectively, in the control group. In both groups, the IADLs scores were higher after the intervention (both time points) than before the intervention, with significant differences (both *p* < 0.05) (Table 2). The GEE approach was used for the difference in IADLs scores before and after the interventions in the two groups, and no significant difference was noted (*p* = 0.325), but a significant difference was noted with regard to the time effect (*p* < 0.001). Thus, the mean IADLs score increased significantly with time. Overall, the results indicated that the MI intervention did not have an impact on BI and IADLs scores among stroke patients.

### 3.3. Changes in Rehabilitation Motivation of Stroke Patients after the MI Intervention

The rehabilitation motivation scale was used for the measurement of the rehabilitation motivation of the stroke patients. In the experimental group, the repeated measures results showed that the mean (SD) rehabilitation motivation scores before and 6 weeks and 3 months after the MI intervention were 21.00 (3.74), 27.54 (2.59), and 26.79 (4.17) points, respectively. The mean rehabilitation motivation score showed an increasing trend over time, with a significant difference (F = 54.60, *p* < 0.001). The MDs for T2–T1 and T3–T1 indicated significant increases, with values of 6.55 and 5.79 points, respectively (both *p* < 0.05) (Table 2).

In the control group, the repeated measures results showed that the mean (SD) rehabilitation motivation scores before and 6 weeks and 3 months after the control intervention were 21.75 (3.02), 25.20 (4.08), and 25.00 (3.72) points, respectively. The mean rehabilitation motivation score showed an increasing trend over time, with a significant difference (F = 16.24, *p* < 0.05). The MDs for T2–T1 and T3–T1 indicated significant increases, with values of 3.45 and 3.25 points, respectively (both *p* < 0.05) (Table 2).

On comparing the mean rehabilitation motivation score before and 3 months after the interventions in the two groups, it was found that the score increased by 5.79 points in the experimental group and increased by 3.25 points in the control group after the intervention (Figure 2). The GEE approach was used for the difference in rehabilitation motivation scores before and after the interventions in the two groups, and the result before the intervention in the control group was used as the baseline. With control of groups and time, it was found that the rehabilitation motivation score in the experimental group was 0.75 points lower than that in the control group, and the difference was not significant (*p* > 0.05), but a significant difference was noted with regard to the time effect (*p* < 0.001). Thus, the mean rehabilitation motivation score increased significantly with time. With control of the interaction variable, it was found that the rehabilitation motivation score in the experimental group was 2.54 points higher than that in the control group after the intervention, and the difference was significant (*p =* 0.026) (Table 3). Overall, the results indicated that the MI intervention had a positive impact on the rehabilitation motivation score among stroke patients.

## 4. Discussion.

### 4.1. Comparison of Differences in Activities of Daily Living (BI and IADLs) between the Experimental and Control Groups

MI can enhance the intrinsic motivation of stroke patients, and it might be helpful for improving their activities of daily living in specific ways [17,21]. The results of this study showed that the 6-week MI intervention and control intervention both significantly improved the BI and IADLs scores, but the BI and IADLs scores between the two groups did not show significant differences. These results are consistent with the findings of the study on MI intervention conducted by Cheng et al. [21] There are several possible reasons for the absence of significant differences in the present study. First, the timing of the MI intervention and the short period of intervention might have influenced the results. The MI intervention in this study was performed in the daytime or nighttime, and most MI interventions were performed after patient rehabilitation, leading to different levels of physical and mental fatigue, which might have had some impact on the intervention effect [16,21,22]. Furthermore, the duration of the intervention in this study was 6 weeks, with only 15–30 min of MI in each session. The number of intervention sessions might need to be increased to eight, and the duration might need to be prolonged to 45 min to enhance the effectiveness of MI [23]. Second, limb hemiplegia might have had adverse effects in the first-stroke patients. The focus of post-stroke care was recovery of limb function in an attempt to restore physical function as soon as possible to achieve the goal of having independent self-care capabilities [11]. Studies showed that approximately 50% of stroke patients had difficulties performing daily activities in the chronic phase (>6 months) [24], and evidence-based research showed that stroke patients were unable to integrate upper limb function into daily life 6 months after stroke and that the dependence of IADLs was influenced 2–3 years after stroke [4,5]. The participants in this study were first-stroke patients who were recruited within 3 months after the stroke. The MI intervention might improve BI and IADLs scores, but the patients still need continuous physical and functional therapy to help them return to their daily lives and to effectively improve their activities of daily living [24]. Third, the first stroke might have had some psychological impact. Post-stroke depression can affect the motivation for rehabilitation and the performance of activities of daily living [4,10]. It has been shown that approximately one-third of stroke survivors were influenced by post-stroke depression after their first stroke [4,11]. Although MI would be helpful for improving activities of daily living in stroke patients, first-stroke patients might be affected by early psychological problems that require some time for psychological adjustment. Depression might be one of the reasons for the ineffectiveness of the MI intervention in first-stroke patients [20,21].

### 4.2. Comparison of Motivation for Rehabilitation Between the Experimental and Control Groups

Motivation for rehabilitation is very important in stroke patients, and it is mainly associated with the convenience of daily life, work, quality of life, and other such factors [24]. MI enables patients to actively understand information related to the disease and improves their confidence, making them want to try to face and solve the pain or frustration experienced in the rehabilitation process and strengthening their motivation to receive rehabilitation information [5,24]. In the present study, the 6-week MI intervention improved the motivation for rehabilitation among first-stroke patients, and the difference was significant. This result is consistent with the findings of the studies on MI intervention conducted by Anstiss [25], Wagner et al., [26] and Balaam et al. [27] Anstiss [28] used the MI intervention to help patients reconnect with their values and experiences, respond to individual patient preferences, needs, and values, trigger potentials for positive emotions (interest, hope, satisfaction, and inspiration), help patients recall past success, and help patients develop abilities and confidence, enabling patients to enhance their motivation for rehabilitation and improve their lives. Wagner et al. [29] provided information needed in MI to change patient confidence, prepare patients for change, facilitate problem solving, and help patients make the final decision consistent with their intrinsic goals and with rehabilitation. Balaam et al. [27] presented a four-stage course and adopted MI to motivate stroke patients at home. The right contents or activities were identified through a curriculum design, and the understanding of the broad social context was used to balance work, responsibility, and enjoyment to maintain patient motivation for rehabilitation and activeness for participation in specific activities or contents to reach the goals of rehabilitation. However, in the present study, it was found that the rehabilitation motivation score increased with time in the control group. The reason for this could be that during the initial adjustment process, the stroke patients were not able to accept the situation themselves, and they were afraid of becoming a burden to their families, causing large physical and mental impacts, including the loss of body autonomy, which led to dysfunction in activities of daily living. With individual attention and routine care, the nurse established a good therapeutic patient relationship with empathy. The active care, individual attention, and listening skills enabled the stroke patients to face frustration bravely and to learn self-care, triggering motivation for rehabilitation [28,29]. It was shown that with individual attention from a nurse, patient motivation for rehabilitation improved. Additionally, it was confirmed that the MI intervention helped improve motivation for rehabilitation among first-stroke patients, indicating that the MI intervention is indeed a specific approach to enhance the intrinsic motivation of stroke patients.

The following were the limitations of our study. The results confirmed that the MI intervention could be very beneficial for the improvement of rehabilitation motivation and activities of daily living among first-stroke patients. However, patients often lack proper home guidance from the professional medical team after discharge and their daily life functions were seriously impacted. The government has not yet completed the construction of the follow-up services for stroke patients after discharge (such as home physical therapy or some group training provided in the communities), and this is the part that needs to be worked on in terms of the current policy. Convenience sampling was used in the recruitment process in this study without random assignment to avoid the contamination caused by the MI intervention between the experimental group and the control group, and the cluster assignment approach was applied with the recruitment of subjects first in the experimental group and then in the recruitment of subjects in the control group. However, the timing of the recruitment of subjects for intervention was different. Furthermore, the researchers assessed the patients’ stage of behavior change on the basis of the information provided by the patients; however, owing to different stages of behavior change in these patients, the researchers reached different goals of the MI on the basis of the difference in the individuality of these patients. This was an insurmountable limitation in the sampling of the subjects. In addition, owing to limited funding, time, and manpower, a larger effective sample size could not be recruited; therefore, the observations of the representative sample cannot be applicable to all stroke patients. It is suggested that the samples source be enlarged and different hospitals be included for obtaining samples to increase research robustness and generalizability.

Several recommendations are proposed. First, the MI intervention should be incorporated into the development education of professional teams and the professional care capability should be expanded through training courses and practical exercises for application in clinical practice to improve the quality of patient care. Second, the extent of the physical and mental fatigue of patients should be considered in the MI intervention and a specific point during continuous interviewing should be selected as a break for patients so that the information obtained can be closer to real-world information. Third, more rigorous experimental research should be conducted in terms of the research design and the randomized controlled research method should be used for a more empirical basis of care. Fourth, the number of sessions of the MI intervention should be increased and the period of intervention should be prolonged to 6 months or even 1 year after stroke to further explore changes. This will help verify the effectiveness of MI with regard to activities of daily living and motivation for rehabilitation and increase the credibility of the research results.

## 5. Conclusions

The present study found that MI could effectively enhance motivation for rehabilitation among stroke patients, although it might not be better than individual attention from a nurse for improving the performance of activities of daily living. Professional teams should learn more MI techniques and apply them in clinical practice to expand their professional roles and improve the quality of care for stroke patients.

## Figures and Tables

**Figure 1 ijerph-17-02755-f001:**
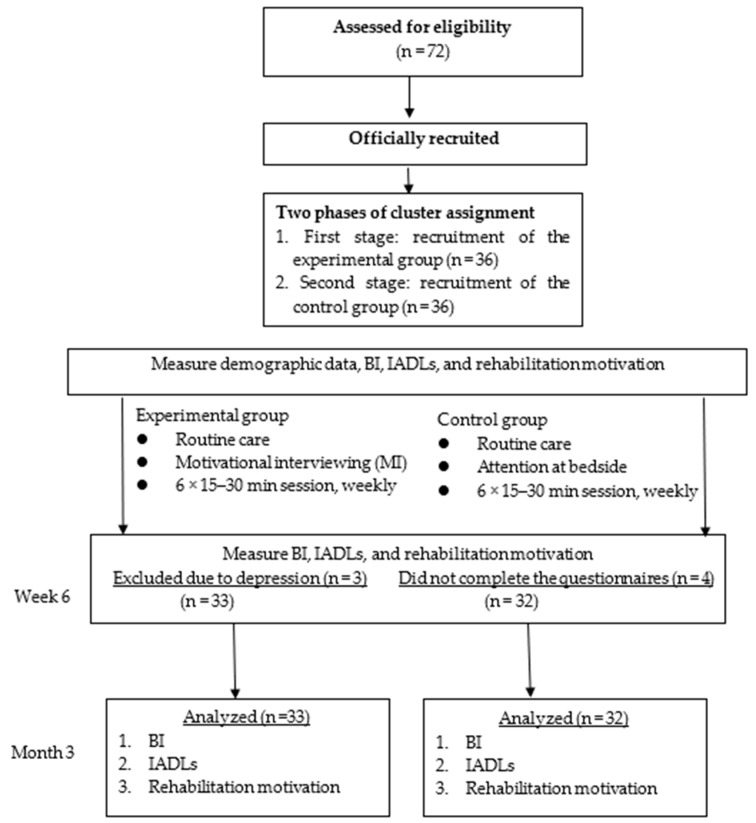
Flow diagram of intervention process through the study; BI: Barthel index, IADLs: instrumental activities of daily living.

**Figure 2 ijerph-17-02755-f002:**
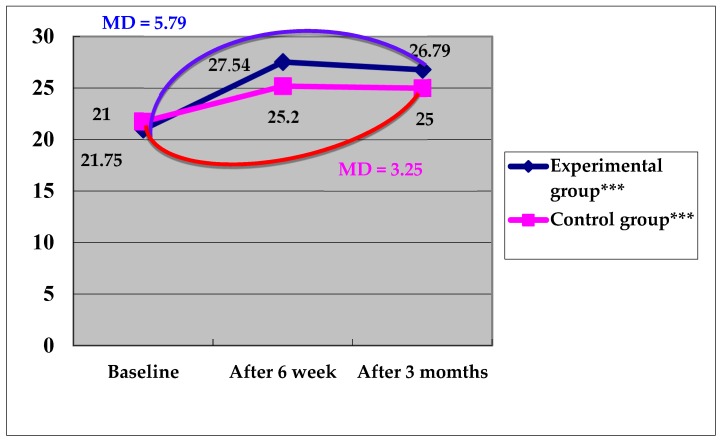
Changes in the rehabilitation motivation scores in the two groups at different testing. ***: The mean rehabilitation motivation score showed an increasing trend over time (6 weeks and 3 months after the MI intervention), with a significant difference *p* < 0.001.

**Table 1 ijerph-17-02755-t001:** Comparison of homogeneity between the two groups in terms of the basic characteristics before the intervention.

Variables	Total (*N* = 65)		Experimental Group (*N* = 33)		Control Group (*N* = 32)			
	*N*	%	*N*	%	*N*	%	*x^2^*	*p*
Sex							0.23	0.639
Male	47	72.3	23	69.7	24	75.0		
Female	18	27.7	10	30.3	8	25.0		
Marital status							0.13	0.727
Unmarried/widowed/divorced	17	26.2	8	24.2	9	28.1		
Married/cohabiting/separated	48	73.8	25	75.8	23	71.9		
Living situation							2.05	0.157
Solitary	5	7.7	1	3.0	4	12.5		
With others	60	92.3	32	97.0	28	87.5		
Religion							1.97	0.168
No	15	23.1	10	30.3	5	15.6		
Yes	50	76.9	23	69.7	27	84.4		
Level of education							12.45	0.060
Junior high school/under junior high school	37	56.9	25	75.8	12	37.5		
High school (vocational)	15	23.1	5	15.2	10	31.3		
College or above	13	20.0	3	9.1	10	31.3		
Working situation							1.06	0.312
No	56	86.2	27	81.8	29	90.6		
Yes	9	13.8	6	18.2	3	9.4		
Economic conditions								
Main source of income							2.09	0.470
Children/spouse/brothers or sisters/parent/other	34	52.3	19	57.6	15	46.9		
Pension/government grants	16	24.6	6	18.2	10	31.3		
Work	15	23.1	8	24.2	7	21.9		
Adequacy of the cost of living							6.60	0.776
Sufficient and more than/roughly enough	31	47.7	11	33.3	20	62.5		
Inadequate	32	49.2	20	60.6	12	37.5		
Very inadequate	2	3.1	2	6.1	0	0		
Smoking history							0.02	0.892
Non-smoker	36	55.4	18	5.4	18	56.3		
Smoker	29	44.6	15	45.5	14	43.7	0.38	0.544
Type of stroke								
Blockage/ischemic stroke	33	50.8	18	54.5	15	46.9		
Hemorrhagic stroke	32	49.2	15	45.5	17	53.1		
Stroke area							9.35	0.543
Anterior cerebral artery	12	18.5	7	21.2	5	15.6		
Middle cerebral artery	21	32.3	12	36.4	9	28.1		
Posterior cerebral artery	8	12.3	6	18.2	2	6.3		
Basal ganglia	10	15.4	2	6.1	8	25.0		
Thalamus	2	3.1	1	3.0	1	3.1		
Intracranial hemorrhage	8	12.3	2	6.1	6	18.8		
Brain stem	4	6.2	3	9.1	1	3.1		
Hemi paralysis							0.12	0.731
Left side	44	67.7	23	69.7	21	65.6		
Right side	21	32.3	10	30.3	11	34.4		
Risk factors for stroke							0.83	0.569
No	4	6.2	2	6.1	2	6.3		
Hypertension	32	49.2	17	51.5	15	46.9		
Diabetes	3	4.6	2	6.1	1	3.1		
Heart disease	4	6.2	2	6.1	2	6.3		
Cardiovascular disease and diabetes	19	29.2	9	27.3	10	31.3		
Hypertension and heart disease	3	4.6	1	3.0	2	6.3		
	Mean	SD	Mean	SD	Mean	SD	*t*	*p*
Age	61.46	13.88	61.61	14.71	61.31	2.60	0.09	0.933
Number of people living together	3.37	2.53	4.12	2.76	2.60	2.05	2.53	0.360
Number of diseases	2.79	1.10	2.33	0.89	3.19	1.26	−3.39	0.052

**Table 2 ijerph-17-02755-t002:** Differences in the effectiveness of the interventions with regard to the BI, IADLs, and rehabilitation motivation scores among the stroke patients in the experimental and control groups at different testing times (*N* = 65).

Items	Mean (SD)			MD		*F*	*p*
	Time 1	Time 2	Time 3	T2 − T1	T3 − T1		
Experimental group (N = 33)							
BI	32.66 (16.66)	51.20 (20.67)	59.38 (23.51)	18.59 *	26.72 *	42.90	<0.001 ***
IADLs	4.38 (3.21)	6.84 (3.82)	7.56 (3.78)	2.47 *	3.19b *	22.88	<0.001 ***
Motivation for rehabilitation	21.00 (3.74)	27.54 (2.59)	26.79 (4.17)	6.55 *	5.79 *	54.60	<0.001 ***
Control group (*N* = 32)							
BI	40.45 (17.96)	58.03 (21.50)	63.15 (26.63)	17.58 *	22.70 *	30.85	<0.001 ***
IADLs	5.24 (2.70)	8.48 (3.95)	9.39 (5.03)	3.24 *	4.15 *	18.10	<0.001 ***
Motivation for rehabilitation	21.75 (3.02)	25.20 (4.18)	25.00 (3.72)	3.45 *	3.25 *	16.24	<0.001 ***

BI: Barthel index, IADLs: instrumental activities of daily living, Time 1 (T1): before the intervention, Time 2 (T2): 6 weeks after the intervention, Time 3 (T3): 3 months after the intervention. * *p* < 0.05, *** *p* < 0.001

**Table 3 ijerph-17-02755-t003:** Comparisons of differences in BI, IADLs, and rehabilitation motivation scores between the two groups at different times using the generalized estimating equation approach.

Parameter	Estimate (B)	S.E.	Wald *x^2^*	*p*-Value
BI				
Intercept	32.66	2.90	126.99	<0.001 ***
Group	−7.80	4.23	3.40	0.070
Time 2 (T2)	17.58	3.00	38.43	<0.001 ***
Time 3 (T3)	22.70	3.52	57.65	<0.001 ***
Group*Time (T2)	1.02	3.61	0.08	0.782
Group*Time (T3)	4.02	5.08	6.26	0.436
IADLs ^c^				
Intercept	4.37	0.56	61.35	<0.001 ***
Group	−0.87	0.72	1.43	0.231
Time 2 (T2)	3.24	0.54	20.81	<0.001 ***
Time 3 (T3)	4.15	0.57	31.01	<0.001 ***
Group*Time (T2)	−0.77	0.78	0.97	0.324
Group*Time (T3)	−0.96	0.98	0.97	0.325
Motivation for rehabilitation				
Intercept	21.75	0.52	1717.79	<0.001 ***
Group	−0.75	0.83	0.82	0.372
Time 2 (T2)	3.45	0.60	33.43	<0.001 ***
Time 3 (T3)	3.25	0.71	20.72	<0.001 ***
Group*Time (T2)	3.10	0.83	14.03	<0.001 ***
Group*Time (T3)	2.54	1.09	5.38	0.026 *

Reference group: control group, reference group: Time 1 (T1, before the intervention), reference group: control group* Time 1, Time 2 (T2): 6 weeks after the intervention, Time 3 (T3): 3 months after the intervention, BI^c^: Barthel index, IADLs^c^: instrumental activities of daily living, SE (standard error) is the standard deviation of its sampling distribution, The Wald *x**^2^*** (also called the Wald Chi-Squared Test) is a way to find out if explanatory variables in a model are significant. * *p* < 0.05, *** *p* < 0.001.

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
