# Peer review of "Effectiveness of Motivational Interviewing in Regard to Activities of Daily Living and Motivation for Rehabilitation among Stroke Patients"

_ijerph, 2020, doi:10.3390/ijerph17082755_

Round 1

Reviewer 1 Report

This is a very well written manuscript.

Only some minor corrections suggestions (often related to technical/language style issues):

Abstract:

Remove the numbers before the abstract subsections

Introduction:

  • line 39: no need to repeat twice ‘activities of daily leaving’
  • lines 43-44: ‘and only’ used repeatedly

Methods

  • Figure 1 presents rather the intervention process – the whole study design - not the participants It might be better to simplify it (the current version do not fit on the page) and be refer to it in section 2.1?

Results

  • Table 1 – title, ‘basic characteristics’ not information?
  • Lines 213-214: these are Methods not Results
  • Table 2 – must be formatted, in current version it is confusing which value refers to which verse; unnecessary dot in the title
  • Table 3 – in previous ones the asterisk is used for p-values, but not here. Suggest to standardize though the whole manuscript
  • Line 247: “Repeated measures ANOVA was used for analysis in both groups” – this are Methods not Results

Discussion

  • The first paragraph of the discussion should be summary of the results
  • Lines 331 and 334 – what are the numbers 34, 10,34?
  • The study implications might be restructured into those for researchers sand practitioners

Author Response

Response to Reviewer 1 Comments

Date: April 12, 2020

Reply to the comments on ijerph-769735

Dear Reviewer:
Thank you for your kind letter of April 12, 2020. We find your comments very important. We believe that your suggestions and editing will greatly improve the quality of our paper. We have, hence, revised our manuscript as you advised and marked our corrections in red and underline type and revised the manuscript in accordance with the reviewers’ comments, and carefully proof-read the manuscript to minimize typographical, grammatical, and bibliographical errors.

(Please see the attachment)

1.Abstract: Remove the numbers before the abstract subsections.

Many thanks for the suggestion. We have removed the numbers before the abstract subsections as advised. Please see Line 15.  

 2. Introduction:

    (1)line 39: no need to repeat twice ‘activities of daily living’

    (2)lines 43-44: ‘and only’ used repeatedly

Thanks for pointing it out. (1) We have deleted the repeated part of ‘activities of daily living’ in Line 39 as advised (Please see Line 39 on Page 1). (2) The term ‘only’ in Line 44 was also deleted (Please see Line 44 on Page 2).

3.Methods:

Figure 1 presents rather the intervention process – the whole study design - not the participants It might be better to simplify it (the current version do not fit on the page) and be refer to it in section 2.1?

Thank you very much for the suggestion. Figure 1 was simplified and referred to in Section 2.1 Study Design as the Reviewer suggested. Please refer to Line 82-83 on Page 2.

4.Results

(1) Table 1 – title, ‘basic characteristics’ not information?

(2) Lines 213-214: these are Methods not Results

(3) Table 2 – must be formatted, in current version it is confusing which value

     refers to which verse; unnecessary dot in the title

(4)Table 3 – in previous ones the asterisk is used for p-values, but not here.

    Suggest to standardize though the whole manuscript

(5)Line 247: “Repeated measures ANOVA was used for analysis in both groups”

    – this are Methods not Results

Thank you very much for the suggestion.

(1)Table 1 – the title was modified from ‘basic information’ to ‘basic

characteristics’. (Please refer to Line 214 on Page 7).

(2)The part describing Methods in Section 3.2 was deleted as the Reviewer advised. Please see Line 220-222 on Page 8.

(3)Table 2 was formatted as the Reviewer suggested. The variables in the table were aligned with the values, and the unnecessary dot in the title in the title was also deleted. Please see Table 2. Line 280 on Page 9.

(4)As the Reviewer advised, the tables in the whole manuscript were standardized while the asterisk was used for p-values of variables with significant difference (*p < 0.05, **p < 0.01, ***p < 0.001). Please see Table 3. Line 286 on Page 10.

(5)The sentence in Line 247 in Section 3.3, “Repeated measures ANOVA was used for analysis in both groups” was deleted. (Please refer to Line 254 on Page 9.)

5.Discussion

(1) The first paragraph of the discussion should be summary of the results

(2)Lines 331 and 334 – what are the numbers 34, 10,34?

(3)The study implications might be restructured into those for researchers and

    practitioners

Thanks for pointing it out.

(1) We have deleted most of unnecessary descriptions. Please refer to Line 300-303 on Page 11 of the manuscript.

(2) Thank the Reviewer for pointing it out. The numbers shown in the manuscript were the literature numbers and were modified after examination. Please refer to Line 335 and 338 on Page 12 of the manuscript.

(3) It is hoped that the results of the research can be interconnected with clinical practice to solve the problems encountered by practitioners. The research results showed that the use of motivational interviewing in first-stroke patients could improve the patients’ activities of daily living [Barthel index (BI) and instrumental activities of daily living (IADLs)], and rehabilitation motivation. Therefore, the restructuring of researchers and practitioners is conducive to the extension of the motivational interviewing intervention to clinical practices, guiding the patients to actively learn the IADLs and participate in rehabilitation so as to improve their quality of life. It is described in Line 385-388 on Page 13 that MI (motivational interviewing) can be included in the development education of professional teams which can enhance the care competency of professionals and also improve the patients’ quality of life.  

Thank you and the reviewer for the kind advice.
Sincerely yours,

Reviewer 2 Report

The paper is well structured

Could there be a change in the results of the last data acquisition because of being made by phone? The other two data acquisition, how were they done? This aspect needs to be clarified.

Has there been any evaluation of MI by EG subjects?

Results

Review table 2, EG (N=3), values are shifted

In discussion section you suggest that the "Depression might be one of the reasons for the ineffectiveness of the MI intervention in first-stroke patients (pag 12, line 327-328)" but also reports that the GE 3 subjects were excluded by depression. Was there any mechanism to detect it?  and if so, I understand that this sentence should not be used. Clarification is needed.

Conclusions

It would be necessary to indicate a conclusion regarding MI and motor performance (the performance of activities of daily living) because it is one of the objectives. To analyze this issue in limitations section is necessary

Author Response

Response to Reviewer 2 Comments

Date: April 12, 2020

Reply to the comments on ijerph-769735

Dear Reviewer:
Thank you for your kind letter of April 12, 2020. We find your comments very important. We believe that your suggestions and editing will greatly improve the quality of our paper. We have, hence, revised our manuscript as you advised and marked our corrections in red and underline type and revised the manuscript in accordance with the reviewers’ comments, and carefully proof-read the manuscript to minimize typographical, grammatical, and bibliographical errors.
(Please see the attachment)

  1. Could there be a change in the results of the last data acquisition because of being made by phone? The other two data acquisition, how were they done? This aspect needs to be clarified.

Thank you very much for the suggestion. Data were collected three times in both the experimental and control groups, including a pre-test in both groups before the intervention of MI, the first post-test 6 weeks after the intervention and the second post-test 3 months after the intervention. The research assistants conducted face-to-face questionnaire survey in the pre-test and the first post-test 6 weeks after the intervention in both the experimental and control groups. The collection of data 3 months after the intervention was conducted among the discharged patients using the same questionnaire via telephone by a trained research assistant who was supervised by the researcher.

   When the patients underwent the telephone survey, errors often occurred because the context could not be standardized, and the non-verbal behavior could not be observed. However, the telephone survey was conducted by a trained research assistant who was supervised by the researcher. Therefore, accuracy of the results of the collected data through telephone could be assured and the results were not affected. Please refer to Line 172-180 on Page 4 of the manuscript.

  1. Has there been any evaluation of MI by EG subjects?

Thanks for pointing it out. Evaluation of MI was conducted  once a week for six weeks among the stroke patients in the experimental group based on the MI Assessment Form and in accordance with the operational manual for the motivational interview for stroke patients for the assessment of motivation for change. Please refer to Line 112-115 on Page 3 of the manuscript.

  1. Results: Review table 2, EG (N=3), values are shifted

Thanks for pointing it out. As the Reviewer suggested, Table 2 was adjusted. Please refer to Table 2 on Page 9. 

  1. In discussion section you suggest that the "Depression might be one of the reasons for the ineffectiveness of the MI intervention in first-stroke patients (pag 12, line 327-328)" but also reports that the GE 3 subjects were excluded by depression. Was there any mechanism to detect it? and if so, I understand that this sentence should not be used. Clarification is needed.

Thank you very much for the suggestion. The depression test among the research subjects was based on the Center for Epidemiologic Studies Depression Scale (CES-D-10), with the full score of 30 points. Depression can be defined when CESD≧10. Therefore, it was mentioned in the Discussion section that depression might be one of the reasons for the ineffectiveness of the MI intervention in first-stroke patients [20,21]. Please refer to Line 102-105 and 331-332 on Page 3, 12 of the manuscript.

  1. Litman T.J. An analysis of the sociologic factors affecting the rehabilitation of physically handicapped patients. Arch. Phys. Med. Rehabil. 1964, 45, 9–16.
  1. Cheng, D.; Qu, Z.; Huang, J.; Xiao, Y.; Luo, H.; Wang, J. Motivational

        interviewing for improving recovery after stroke. Cochrane Database Syst.          Rev. 2015, (6). 1–23, doi:10.1002/14651858.

     5.Conclusions

It would be necessary to indicate a conclusion regarding MI and motor performance (the performance of activities of daily living) because it is one of the objectives. To analyze this issue in limitations section is necessary

Thank you very much for the suggestion. It was the conclusion of this study that Motivational Interviewing could effectively improve the rehabilitation motivation and the performance of activities of daily living among stroke patients. The analysis of these issues was made in the section of Limitations of the Study. Please refer to Line 102-105 and 365-371 on Page 12 of the manuscript.

Thank you and the reviewer for the kind advice.
Sincerely yours